# Bonactin and Feigrisolide C Inhibit *Magnaporthe oryzae Triticum* Fungus and Control Wheat Blast Disease

**DOI:** 10.3390/plants11162108

**Published:** 2022-08-12

**Authors:** S. M. Fajle Rabby, Moutoshi Chakraborty, Dipali Rani Gupta, Mahfuzur Rahman, Sanjoy Kumar Paul, Nur Uddin Mahmud, Abdullah Al Mahbub Rahat, Ljupcho Jankuloski, Tofazzal Islam

**Affiliations:** 1Institute of Biotechnology & Genetic Engineering (IBGE), Bangabandhu Sheikh Mujibur Rahman Agricultural University, Gazipur 1706, Bangladesh; 2Extension Service, Davis College of Agriculture, West Virginia University, Morgantown, WV 26506, USA; 3Plant Breeding and Genetics Section, Joint FAO/IAEA Centre, International Atomic Energy Agency, 1400 Vienna, Austria

**Keywords:** antifungal secondary metabolites, biocontrol, abnormal germ tube suppression of appressoria, *Streptomyces* sp.

## Abstract

Wheat blast caused by the *Magnaporthe oryzae*
*Triticum* (MoT) pathotype is one of the most damaging fungal diseases of wheat. During the screening of novel bioactive secondary metabolites, we observed two marine secondary metabolites, bonactin and feigrisolide C, extracted from the marine bacteria *Streptomyces* spp. (Act 8970 and ACT 7619), remarkably inhibited the hyphal growth of an MoT isolate BTJP 4 (5) in vitro. In a further study, we found that bonactin and feigrisolide C reduced the mycelial growth of this highly pathogenic isolate in a dose-dependent manner. Bonactin inhibited the mycelial development of BTJP 4 (5) more effectively than feigrisolide C, with minimal concentrations for inhibition being 0.005 and 0.025 µg/disk, respectively. In a potato dextrose agar (PDA) medium, these marine natural products greatly reduced conidia production in the mycelia. Further bioassays demonstrated that these secondary metabolites could inhibit the MoT conidia germination, triggered lysis, or conidia germinated with abnormally long branched germ tubes that formed atypical appressoria (low melanization) of BTJP 4 (5). Application of these natural products in a field experiment significantly protected wheat from blast disease and increased grain yield compared to the untreated control. As far as we are aware, this is the first report of bonactin and feigrisolide C that inhibited mycelial development, conidia production, conidial germination, and morphological modifications in the germinated conidia of an MoT isolate and suppressed wheat blast disease in vivo. To recommend these compounds as lead compounds or biopesticides for managing wheat blast, more research is needed with additional MoT isolates to identify their exact mode of action and efficacy of disease control in diverse field conditions.

## 1. Introduction

Wheat is an essential staple dietary source for approximately 2.5 billion individuals in 89 different nations in the world. In low- and middle-income nations, it outperforms maize or rice as a source of protein. Wheat ranks second only to rice in the context of calorie supply. It is a primary food source in North Africa and West and Central Asia, accounting for up to half of the calories consumed (https://wheat.org/; accessed on 16 May 2022). Nonetheless, wheat is prone to various fungal diseases; the most notorious one is a wheat blast, caused by the pathogenic filamentous fungus *Magnaporthe oryzae Triticum* (MoT) pathotype. In 1985, the first case of the wheat blast was recorded in Brazil [1,2]. In 2016, Bangladesh experienced an alarming epidemic of the wheat blast that was the first incidence of the disease in Asia [3]. That epidemic destroyed 15,000 hectares of wheat fields with a yield loss of up to 100% [4]. Wheat blast is causing concerns among seed scientists as it has the potential to spread to important wheat-growing areas in South Asian and African countries [5]. Plant pathologists have warned that the disease might spread to India, Pakistan, and China, which are the second-, seventh-, and first-highest wheat producers in the world, respectively [4,6,7,8].

The MoT is a filamentous haploid ascomycete fungus. Its infection cycle has previously been described [9,10]. Briefly, MoT’s three-celled hyaline airborne conidium lands on a wheat leaf and attaches to it using adhesive. It then begins to grow, developing into a slender germ tube with an appressorium at the tip. A tiny penetrating peg develops at the base of the appressorium, compressing the cuticle and allowing entry into the wheat epidermis. Wheat plasma membranes are penetrated by bulky, virulent mycelium, which then enters epidermal cells to complete tissue invasion [10,11,12]. It affects the aerial parts of the wheat plant, specifically the leaves, stems, nodes, and kernels encompassing all growth phases [7,13,14]. MoT usually affects spikes and bleaches the infected spikes, which results in malformed grains or producing no grain at all [4,15]. Wheat heads with severe infection may die, resulting in a considerable decrease in productivity. The early bleaching of spikelets above the infection point and the whole panicle is the most common symptom [4,7,16]. Contaminated seeds or grains as well as airborne conidia spread this disease, and the pathogen may persist in infected crop residues and seeds [17].

There is an ongoing demand for new plant chemotherapeutic agents that are unique from frequently used fungicides in their underlying mechanisms for advanced plant disease management. Another important reason for these needs is the occurrence of fungicide-resistant pathogens, which results from the requirement of using many synthetic fungicides at high rates, with adverse environmental repercussions [18,19]. Several microorganisms have been authorized as biocontrol agents in many countries including the EU to date due to their relatively low toxic residues, environment-friendly properties, and low manufacturing cost [20]. However, scientific research suggests that these benefits are not always achieved as biological pesticides are mostly living organisms, and their performance varies owing to the influence of numerous biotic (nutritional requirements, host species, and pathogenic microbes) and abiotic (moisture, temperature, relative humidity) factors, which limit their fitness under field conditions [21,22]. In addition, some biological control microbes, such as *Bacillus cereus*, are known to cause human diseases, precluding their release in the environment. In this regard, microbial metabolites can be another suitable alternative to live microbes or synthetic fungicides that are also capable of controlling plant diseases with low detrimental effects on human health and the environment [23]. The versatility of biological activity and chemical structure of microbial metabolites as a pesticide is worth considering due to the potential benefits [24]. The second aspect of microbial metabolites as agricultural fungicides is the requirement of a relatively short period for biodegradation. According to Tanaka and Omura [25], they often decay within a month or even a few days, leaving low residue that should be less harmful to the environment. Metabolites derived from diverse microorganisms have been utilized extensively to address commercially important diseases of several plants [26].

Secondary metabolites extracted from *Streptomyces* species have shown a broad range of biological functions by blocking particular enzymes or proteins in signaling cascades [27,28,29,30]. Wheat blast management research of our working group took a comprehensive strategy including biologicals and biorational approaches. During the screening of new bioactive natural products against MoT, we discovered that a few metabolites of *Streptomyces* spp. inhibited the growth of MoT mycelia [31]. Two natural secondary metabolites, bonactin and feigrisolide C, extracted from marine *Streptomyces* spp., Act 8970 and ACT 7619, respectively, exhibited substantial growth inhibitory effects against a MoT isolate among many different compounds tested. The first acyclic ester of nonactic acid is bonactin, whereas feigrisolide C is a non-symmetric lactone associated with the nactic acid group [32,33]. Bonactin has shown antimicrobial properties against both bacteria and fungi. Many different microbes including *Bacillus megaterium*, *Klebsiella pneumoniea*, *Escherichia coli*, *Micrococcus luteus*, *Staphylococcus aureus*, *Saccharomyces cerevisiae*, and *Alicagenes faecalis* are sensitive to bonactin [33]. In antiviral, antibacterial, and enzyme inhibition tests, feigrisolides were found to have varying degrees of effectiveness. Synthesis of feigrisolide C has been achieved [34]. Nonactic acid esters are in general environmentally benign since soil microbes convert them to H_2_O and CO_2_ [35]. Inhibitory effects of bonactin and feigrisolide C on zoosporogenesis and motility of phytopathogenic Peronosporomycete zoospores have been reported [29]. A few studies have documented the toxicity level of these compounds to date. Bonactin is reported as non-carcinogenic and non-toxic to aquatic model organisms. It has been reported as a suitable natural compound for schizophrenia disorder, suggesting little or no toxicity to humans [36]. However, further research is needed to ascertain their safety for humans and the environment before using them as a potential lead component for the synthesis of agricultural fungicides for controlling wheat blast. There is currently no information available about the use of nonactic acid esters’ antimicrobial activities to control wheat blast disease. To our best knowledge, this is the first report of marine natural antibiotics bonactin and feigrisolide C from *Streptomyces* spp. inhibiting a destructive wheat blast causing a MoT isolate and suppressing the disease in field conditions. The major targets of the current study were to: (i) assess the inhibitory effects of bonactin and feigrisolide C on the mycelia growth of BTJP 4 (5); (ii) evaluate the influences of these marine natural products on conidia production, germination, and the developmental transitions of conidia of BTJP 4 (5); (iii) assess the effect of these compounds on the suppression of wheat blast disease development caused by BTJP 4 (5) on leaves and spikes; and (iv) compare the disease inhibition efficiencies of these natural compounds with a commercialized fungicide Nativo^®^75WG.

## 2. Results

### 2.1. Mycelial Growth Inhibition and Morphological Alteration of Hyphae

Both bonactin and feigrisolide C considerably inhibited MoT mycelium development in the PDA medium in a dose-dependent manner (Figure 1). Bonactin inhibited mycelium development of the MoT isolate BTJP 4 (5) more efficiently than feigrisolide C. When bonactin and feigrisolide C were applied separately at 2 µg/disk, mycelial growth inhibition was 70.8 ± 0.8% and 68.1 ± 1.0%, respectively (Figure 2). Both bonactin and feigrisolide C demonstrated slightly lower inhibitory capacity than Nativo^®^ WG 75 (82.7 ± 0.6% at 2 µg/disk). The inhibitory effects of these natural compounds enhanced as concentrations were raised from 0.005 to 2 μg/disk, reaching up to 71% for bonactin (Figure 2). Bonactin had more inhibitory efficacy than feigrisolide C but was slightly less effective than Nativo^®^WG 75 against the BTJP 4 (5) isolate. Both substances were ineffective against MoT at quantities lower than 0.005 μg.

Bonactin extensively impeded BTJP 4 (5) hyphal growth at 2 μg/disk (70.8 ± 0.8%), 1.5 μg/disk (65.4 ± 1.0%), and 1 μg/disk (58.6 ± 1.3%), showing that inhibition and accelerated concentrations had a positive correlation. At 2, 1.5, and 1 μg/disk, feigrisolide C inhibited 68.1 ± 1.0%, 63.4 ± 1.3%, and 54.2 ± 1.0% hyphal growth of BTJP 4 (5). Bonactin and feigrisolide C had minimum suppressive concentrations of 0.005 and 0.025 μg/disk, respectively, and these compounds suppressed mycelial growth by 9.81 ± 1.3% and 13.3 ± 0.8% at 0.005 and 0.025 μg/disk, respectively. However, the minimal inhibitory concentration of Nativo^®^ WG 75 was 0.05 μg/disk, but at higher doses starting at 0.1 μg/disk it outperformed the suppression percentage of the two other test compounds at equal concentrations. It is worth noting that at less than 0.1 μg/disk concentration, bonactin and feigrisolide C inhibited mycelial development more efficiently than Nativo^®^ WG 75, and bonactin inhibited BTJP 4 (5) at a 10-fold lower dose.

Microscopic examinations of untreated BTJP 4 (5) revealed polar, cylindrical growth with smooth, hyaline, branching, plump, septate, and unbroken hyphae (Figure 1a,a′). Hyphae treated with bonactin and feigrisolide C grew irregularly and exhibited a higher frequency of branching per unit of the hyphal length. Cell walls of the hyphae were not smooth but had ridges that gave them a crinkled look as well as causing irregular cell swelling (Figure 1b,b′,c,c′). Similar effects of the fungicide Nativo^®^WG75 on hyphal growth were observed. Mycelia closer to the filter disk of Nativo^®^WG75 showed a comparable modification of MoT hyphae (Figure 1d,d′). However, compared to Nativo^®^WG75, the two natural products generated slightly different morphological aberrations in MoT, suggesting a possibly different mode of action.

### 2.2. Conidiogenesis Inhibition

Bonactin, feigrisolide C, and Nativo^®^WG75 considerably decreased the conidia production of BTJP 4 (5) at concentrations of 1, 5, and 10 µg/mL, respectively, and suppression increased with increasing concentrations from 1 to 5 to 10 g/mL (Figure 3). Almost no or only a few conidia were produced at 10 µg/mL in media amended separately with all three compounds. Microscopic examination also revealed broken hyphal tips and complete suppression of conidiophore formation in fungal colonies in Petri plates that were treated with these three compounds at 10 µg/mL.

### 2.3. Inhibition of Conidia Germination and Morphological Aberrations in Germinated Conidia

To determine the MoT isolate BTJP 4(5)’s conidial germination inhibition capacity of test products, bonactin, feigrisolide C, and Nativo^®^ WG75 were added to the multi-well plates at concentrations of 0.5 µg/mL. The rate of conidial germination was recorded after 6, 12, and 24 h of incubation at 25 °C (Table 1). Bonactin and Nativo^®^ WG75 treatments dramatically inhibited conidia germination compared to the control, while no conidia germinated in feigrisolide C-treated plates after 6 h of incubation. All (100%) conidia germinated in the water while it was 49.7 ± 0.6% in Nativo^®^ WG75-treated plates. The BTJP 4 (5)’ conidia germination rates with bonactin and feigrisolide C were 79.1 ± 0.6% and 0 ± 0%, respectively, at 0.5 μg/mL.

In the dark at 25 °C, 100% of conidia germination occurred in water during all incubation periods (6 h, 12 h, and 24 h), with normal germ tube and mycelial growth (Table 1, Figure 4a). At 0.5 µg/mL, both bonactin (panel b) and feigrisolide C reduced on the germination of the conidia and the post-germination developmental processes, resulting in abnormal transitions from one stage to another. During 6 h of incubation in the presence of bonactin, the conidia germination rate was 79.1 ± 0.6%, which had short germ tubes. After 12 h, 12.7 ± 0.4% of normal germ tubes were observed, whereas 66.5 ± 0.5% had abnormally long branched germ tubes. After 24 h, there were 9.5 ± 0.2% normal appressoria and 60.1 ± 0.3% atypical appressoria (low melanization), without any hyphal development (Table 1, Figure 4b).

In the case of feigrisolide C, 7.4 ± 0.5% of the conidia lysed after 6 h, and no germination occurred between 6 h and 24 h (Table 1, Figure 5c). In the presence of Nativo^®^ WG75, 49.7 ± 0.6% of conidia germinated with normal germ tubes after 6 and 12 h, but no appressorial development took place. Nativo^®^ WG75 also inhibited sporulation similar to feigrisolide C to prevent further mycelial growth after 24 h (Table 1, Figure 5A). It is worth mentioning that these compounds resulted in excessively long branching in germ tubes and conidia lysis, whereas Nativo^®^ WG75 had no such effect.

### 2.4. Wheat Blast Progression on Excised Wheat Leaves

The two metabolites applied at 1, 5, and 10 µg/mL considerably decreased the wheat blast disease symptoms in excised leaves of wheat infected with BTJP 4 (5). The lesion lengths in the leaves pretreated with bonactin were 5.3 ± 0.2 mm at 1 µg/mL and 1.2 ± 0.2 mm at 5 µg/mL, respectively (Figure 5A,B). The blast lesion lengths with feigrisolide C were 6.1 ± 0.2 mm and 2.5 ± 0.1 mm at 1 µg/mL and 5 µg/mL, respectively (Figure 5A,B). Leaves of wheat treated with bonactin, feigrisolide C, and Nativo^®^WG75 at 10 µg/mL did not show any blast symptoms (Figure 5A,B). Normal blast lesions were visible on the water-treated leaves with average lengths of 9.3 ± 0.2 mm (Figure 5A,B). In comparison to both compounds, the fungicide effectively reduced lesion progression at 1 and 5 µg/mL.

### 2.5. Wheat Blast Disease Suppression in the Field at the Heading Stage 

To determine the efficacy of these compounds in suppressing blast disease in artificially infected wheat spikes by BTJP 4 (5), a field experiment was conducted by using a commercial fungicide Nativo^®^75WG at 50 µg/mL as a local standard. In the field, bonactin and feigrisolide C considerably reduced wheat blast disease incidences (41% and 51.3%, respectively) (Figure 6c,d, Table 2), compared to 87.3% disease incidence in the untreated control (Figure 6b, Table 2).

Furthermore, 32.3 ± 2.40% and 38.6 ± 1.20% blast severities were recorded in wheat plants pretreated with these compounds in comparison to 82.6% in the untreated control. Bonactin (112.9 ± 2.26 gm), feigrisolide C (106.4 ± 2.58 gm), and Nativo^®^ 75WG (126.1 ± 2.70 gm) had significantly increased grain yields compared to the untreated control (64.6 ± 1.71 gm). Grain yields in the Nativo^®^ 75WG were statistically similar to the healthy control (133.1 ± 2.33 gm). Nevertheless, both bioactive natural compounds’ treatments had statistically lower but similar grain yields compared to the Nativo^®^ 75WG fungicide and healthy control (Table 2).

Thousand-grain weights for Nativo^®^ 75WG, feigrisolide C, bonactin, and the negative control were 43.2 ± 2.52, 38.7 ± 3.16, 40.1 ± 1.72, and 46.6 ± 1.57 gm, respectively. Grain yields in treated plots were considerably greater than the yield of the untreated control plot (31.7 ± 1.29 gm) (Table 2).

## 3. Discussion

In this study, we demonstrated for the first time that two nonactic acid esters extracted from marine *Streptomyces* spp. and named bonactin and feigrisolide C inhibited the growth and development of a destructive wheat blast pathogen *M. oryzae Triticum* (MoT) isolate BTJP 4 (5). Additionally, we discovered that these natural compounds were comparable to the commercial fungicide Nativo^®^ WG75 in their efficacy in successfully reducing wheat blast disease in wheat leaves and spikes that had been artificially inoculated by BTJP 4 (5). These treatments also resulted in a modest increase in grain yield although the highest yield was obtained from fungicide treatment followed by two test compounds. Formation of conidia asexually in hyphal conidiophore and germination of conidia are critical for plant infection by the blast fungus [37,38,39,40]. Suppression of hyphal growth, conidia formation, and germination of many fungi, such as rice and wheat blast fungi, by various natural products, have been reported [29,31,33,41,42,43,44,45]. The nonactic acid esters are precursors of macrotetrolide antibiotics which have a broad spectrum of antimicrobial, anticancer, acaricidal, insecticidal, immunosuppressive, antiprotozoan (coccidiostatic), and antiparasitic properties [35,46,47,48]. In the current study, we did not focus on unraveling the underlying molecular mechanism associated with in vitro growth inhibition of wheat blast causing fungal pathogen and suppression of the disease in vivo. However, from a similar study, Islam et al. [29] found that the hydrolysis of mitochondrial ATP via increased ATPase function was likely associated with the mode of action of antimicrobial activities of macrotetrolides against phytopathogenic Peronosporomycete zoospores. Despite having outstanding biological properties, macrotetrolides have received extremely less attention in plant protection studies. To the best of our knowledge, it is the first report of two natural bioactive nonactic acid esters and precursors of macrotetrolides (bonactin and feigrisolide C) originated from marine *Streptomyces* spp. suppressing the highly aggressive wheat blast pathogen MoT isolate BTJP 4 (5) in vitro and in vivo. Additional study is needed to test the efficacy of these compounds against other strains of MoT as well as whether their antiblast activities are linked with the induction of increased mitochondrial ATPase activity in the asexual spores and hyphae of MoT. 

One of the key discoveries of this study is that at almost equal concentrations of Nativo^®^ WG75, both bonactin and feigrisolide C dramatically reduced hyphal growth, conidia production, and germination, and also caused morphological changes in germinated conidia. Our findings indicate that these natural substances inhibited conidial germination and mycelium growth, which consequently suppressed wheat blast disease in vivo.

The swelling phenomenon by these compounds on BTJP 4 (5) hyphae is another remarkable observation from our study (Figure 1b’–d’). We utilized doses ranging from 0.005 to 2 µg/disk in our experiment. Swelling increased with increased concentrations, showing a positive correlation of swelling with concentrations. Tensin [49], fengycin [50], gageopeptides, gageotetrin [44], and oligomycins [31] have all been reported to induce developmental aberrations in the tubular growth of the fungal hyphae. Developmental transitions in *Aphanomyces cochlioides* hyphae, such as increased swelling and excessive branching, have been observed in response to xanthobaccin A from *Lysobacter* sp. SB-K88 or m *Pseudomonas fluorescence* phloroglucinols [51,52,53,54]. According to Schumacher et al. [33], bonactin from *Streptomyces* sp. greatly suppressed the hyphal development of *Saccharomyces cerevisiae*, but no data on the mycelial growth inhibitory activity of feigrisolide C has been documented to date. So far, this is known to be the first report of some nonactic acid and nonactic acid ester exhibiting swollen-like abnormal hyphae against a destructive wheat pathogen. 

Conidiogenesis is the process of producing conidia, which are fungal spores that are grown asexually on the conidiophore [39]. The majority of fungal plant pathogens attack plants by these asexual spores. Inhibiting or preventing conidiogenesis and conidia germination can reduce the likelihood of host infection by fungal pathogens [55,56]. Future plant protection strategies should explore and rely on similar natural compounds that interfere with these processes. Therefore, another noteworthy finding from this study was that these compounds greatly decreased conidiogenesis (Figure 3), and conidial germination, and also triggered morphological alterations of BTJP 4 (5)’s conidia (Table 1, Figure 4). 

Lysis of conidia and uneven branching of germ tube tips as well as unusually long hypha-like germ tubes were among the other distinct and interrelated phenomena found in this work (Figure 4B,C). Dame and co-workers [57] discovered a similar occurrence when they found that oligomycins derived from a marine *Streptomyces* sp. triggered lysis of phytopathogenic *Plasmopara viticola* zoospores that causes grapevine downy mildew disease. Homma and colleagues reported that lecithin induced abnormal branching in germ tube tips of rice blast fungus, and prevented the development of appressoria [58]. Similarly, *A. cochlioides*’ cystospores germinated with hyperbranched germ tubes by the effects of diacetylphloroglucinol (DAPG) [54]. Bonactin caused atypical appressoria (low melanization), which restricted MoT fungal infection since appressorium melanization is essential for *M. oryzae* pathogenicity [11]. This compound may affect the gene expression related to the synthesis of melanin. This is also the first study to show that two esters impeded conidiogenesis, germination, and the development of appressoria of BTJP 4 (5) conidia. Future research should concentrate on the mechanisms by which these compounds suppress conidia formation, germination, and appressorium formation of MoT, as well as the impact of these natural bioactive compounds on the expression of genes associated with conidia germination and appressorium formation of BTJP 4 (5) or similar MoT isolates.

Nonactic acid esters are relatively safe for the environment since soil microorganisms can quickly convert them to H_2_O and CO_2_ [59]. Plant growth stimulation and specific insecticidal actions of nonactin antibiotic precursors have been documented [35,60]. Bonactin was reported to have antibacterial action and also antifungal action [33]. In a lab investigation, we noticed that nonactin had remarkable antifungal properties against MoT both in vivo and in vitro (our unpublished data). According to Schumacher et al. [33], antimicrobial activity can be achieved without the requirement for a macrotetrolide ring structure, such as the non-asymmetric lactone feigrisolide C, which has antibacterial and antiviral properties [61]. Islam and his colleagues [29] discovered that bonactin and feigrisolide C with other known macroletrolides suppress zoosporogenesis, hamper motility, as well as trigger lysis of *Plasmopara viticola* zoospores. The findings of the current work do not elucidate the detailed mechanism of action, but they do suggest that stimulation of ATPase activities in mitochondria or/and imbalance/translocation of cell cations could inhibit hyphal development and impede conidia germination. Identifying the role of ATPase in inhibiting hyphal growth, conidiogenesis, conidial germination, and appressoria formation may aid in our understanding of the biology and pathogenesis of filamentous plant pathogens. This naturally occurring ATPase inducer may thus be a promising pioneer ingredient for developing novel, efficient agrochemicals to fight this aggressive fungal pathogen.

In this study, wheat leaves pretreated with the test compounds showed shorter lesions than untreated checks (Figure 5). The majority of those lesions were small, and appeared as brown patches with spots of a pinhead size (scale 1) to roundish and fairly expanded grey dots that ranged in size from 1–2 mm in diameter (scale 3). The untreated control leaves had typical blast lesions covering 26–50% of the leaf surface (scale 7), according to the 9-scale blast disease assessment system developed by the IRRI SES (standard evaluation system) [55,56]. However, in the Nativo^®^ WG75 treatment, no visible blast lesions were present. When disease control studies were conducted at the wheat heading stage, similar results were obtained. In artificially infected wheat spikes, blast disease progression was dramatically inhibited by bonactin and feigrisolide C (Figure 6). A popular systemic fungicide, Nativo^®^ WG75, was used in this study as a local standard and positive control. In terms of suppression efficacy of the MoT fungus, the two marine natural compounds evaluated in the current study were comparable to that of the commercial fungicide. The active components of Nativo^®^ WG75 are tebuconazole and trifloxystrobin. Belonging to the systemic triazole fungicide group, tebuconazole’s mode of action is known as demethylase inhibitor (DMI). The development of the fungus is slowed down and can eventually be killed as DMI fungicides interfere with the production of sterol in fungal cell walls [62]. Trifloxystrobin, a fungicide in the strobilurin group, suppresses the spore germination of phytopathogenic fungi by disrupting energy production through blocking mitochondrial electron transport [62]. The modes of action of bonactin and feigrisolide C are possibly distinct from Nativo^®^WG75, despite the observation of a similar disease inhibition response. More research is needed to determine the fundamental mechanism through which these compounds suppress wheat blast. Before acknowledging these compounds as prospective fungicides for wheat blast, a large-scale field evaluation of their efficiency in preventing wheat blast infection is necessary. Recently, it has been found that secondary metabolites from both marine and terrestrial species can biologically suppress the wheat blast disease [55,56].

In today’s agriculture, the development of fungicide resistance across pathogenic microorganisms is a major concern. Due to the inappropriate use of fungicides with a single-site active mode of action such as triazole and strobilurin (QoI), some resistant MoT mutant species have been found widely distributed [19]. Investigators are actively searching for new, effective antifungal chemicals possessing alternative modes of action to protect wheat plants against this lethal pathogenic fungus due to the risk of resistance development in conventional fungicides. The marine natural products, bonactin and feigrisolide C, exhibited almost equivalent bioactivity to the commercialized fungicide Nativo^®^WG75. The effectiveness of these compounds as inhibitors of the MoT isolate BTJP 4 (5) has suggested using them as candidates for agrochemical with a novel mode of action towards this wheat pathogenic fungus provided they are equally effective on other MoT strains under various agro-ecological regions. Further studies are needed with structurally diverse nactic acids, their esters, and macrotetrolides for understanding the structure-activity relationship of bonactin and feigrisolide C. However, very few reports have been published yet regarding their impacts on humans and the environment, and more research is also required to assess their toxicity level before using them to produce fungicides.

## 4. Materials and Methods

### 4.1. Fungal Isolate, the Revival of a Synthetic Medium, and Host Plant Materials

In 2016, during the first wheat blast epidemic in Jhenaidah, Bangladesh, we collected many MoT strains including BTJP 4 (5) from wheat cv. Prodip (BARI Gom-24) that showed blast infection on spikelets These isolates were preserved at 4 °C on dried filter paper for later use. We revived five isolates on a potato dextrose agar (PDA) medium and tested them in the lab for their normal colony characters and aggressiveness to select a representative one (BTJP 4) for this work (4). We also tested isolates collected from the field-infected wheat from 2016 to 2022 and found that they were equally sensitive to the commercial fungicide, Nativo^®^ WG 75 (Figure 7). It appeared the clonal population introduced in Bangladesh from South America had not been mutated [37]. Therefore, we chose BTJP 4(5) for the whole study. On a potato dextrose agar (PDA) medium, the selected isolate was grown for seven days at 25 °C. Ten-days-old PDA-grown fungi fungal colonies were washed in an aseptic environment in a laminar flow hood with 500 mL of deionized water to remove aerial mycelia, and then kept at ambient temperature (25–30 °C) for 2–3 days to induce abundant conidia production [4,40,55,56]. The conidia were scraped out from each plate using a glass slide after adding 15 mL of water to each plate. Two-layer cheesecloth was used to filter out the hyphal mass, and the dilution was conducted to achieve 1 × 10^5^ conidia/mL. Conidial germination was examined under a compound microscope by counting the number. Seedlings of blast disease-susceptible wheat variety Prodip (BARI Gom-24) at the five-leaf stage were used for the bioassay on leaves [31,55,56]. For assessing the wheat blast disease suppression efficacy of bonactin and feigrisolide C, these compounds were sprayed on field-grown wheat spikes at the flowering stage one day before inoculation of the plants with MoT conidia. The detailed method of artificial inoculation of wheat plants by MoT conidia was described recently by Paul et al. [40].

### 4.2. Chemicals

Bonactin and feigrisolide C (Figure 8) were derived from the marine bacteria *Streptomyces* spp. Act 8970 and ACT 7619. Dr. Hartmut Laatsch, a Professor of Georg-August-Universitaet Goettingen in Germany, generously provided these pure chemicals as gifts [29]. The fungicide Nativo^®^ WG 75 (50:50 mixtures of trifloxystrobin and tebuconazole) was purchased in Dhaka, Bangladesh from Bayer Crop Science Ltd. Stock solutions of test compounds were prepared using small amounts of DMSO (dimethyl sulfoxide), and then the solutions were diluted with water. The final mixture included a maximum of 1% (*v*/*v*) DMSO, which had no impact on the development or sporulation of BTJP 4 (5) mycelium [55,56].

### 4.3. Suppression of Mycelial Growth and Hyphal Morphological Alteration

Using a modified disk diffusion technique as reported by Chakraborty et al. [31], the mycelial growth inhibition of MoT isolate BTJP 4(5) was determined by the application of bonactin, feigrisolide C, and the commercial fungicide Nativo^®^WG75 on filter paper disks. To prepare a range of concentrations from 0.005 to 2 µg/disk, the required amounts of natural compounds and the fungicide Nativo^®^ WG75 were dissolved in ethyl acetate and water. Nine-millimeter-diameter filter paper disks (Sigma-Aldrich Co., St. Louis, MO, USA) were used to absorb the test compound solutions. In 9 cm-diameter Petri dishes with 10 mL of PDA, the treated disks were placed 2 cm apart from one side. The filter paper disks containing the test chemicals were placed on the opposite side of the actively growing 5 mm-diameter, 7-days-old mycelial plugs of BTJP 4(5). Petri dishes with fungal hyphal plugs against filter paper disks with Nativo^®^WG75 were used as a control. As a negative control, filter paper disks were coated with ethyl acetate and then allowed to evaporate at ambient room temperature. A fungal hyphal development reduction was observed after 10 days of culture. The Petri plates used as untreated controls were incubated at 25 °C until the fungus had colonized and covered the whole surface of the agar. The test was conducted five times with five replications for each concentration. Using a ruler and two perpendicular lines drawn on the lower side of each plate, the radial growth of the fungal culture was measured in centimeters. Measurements were also recorded for the inhibition zone and associated fungal colony diameter influenced by the test compounds and the fungicide. Inhibition percentage radial growth (RGIP) [55,56] was calculated as:
RGIP % =Control plate radial growth − Treated plate radial growth× 100
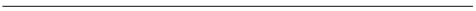
Control plate radial growth

Results including radial growth suppression from the disk diffusion test were captured using a digital camera of CAMEDIA C-3040 zoom. At 40× and 100× magnification, an Olympus IX70-S1F2 microscope was used to study the mycelial morphology at the sharp end of the cultures approaching the control and treated disks. The mycelial growth including aberration was photographed using the same digital camera attached to the microscope.

### 4.4. Suppression of Conidiogenesis

The stock solutions of each compound were prepared in 10 μL of DMSO and then diluted with distillate water to obtain concentrations of 1, 5, and 10 μg/ml. The final mixtures had a maximum of 1% (*v*/*v*) of DMSO, which had no impact on BTJP 4 (5) sporulation or hyphal development. A 5 mL solution of Nativo^®^WG75 was prepared to achieve each 1, 5, and 10 μg/mL concentrations by dissolving the required amount of formulation in distilled water that was used as a positive control. A conidiogenesis inhibition test of a MoT isolate was established in our lab and used for this work [31,39,56]. Briefly, to deplete nutrients and promote conidiogenesis, the mycelium of a 10-day-old BTJP 4 (5) Petri plate culture was rinsed [4,39]. After being treated, ten mm BTJP 4 (5) hyphal agar blocks were treated with 50 µL of each test compound and Nativo^®^WG75 at the aforementioned doses and then placed on Nunc multi-well plates. The mycelial agar block of MoT with 1% DMSO in the same amount of sterile water was used as a negative control. Treated BTJP 4 (5) mycelial plugs were incubated at 28 °C and >90% RH under alternating light and dark cycles for 14 and 10 h, respectively. After 24 h, conidiogenesis was observed under a 40× Zeiss Primo Star microscope for analysis, and pictures were taken with a Zeiss Axiocam ERc 5s. With five replications for each treatment, the test was repeated five times.

### 4.5. Suppression of Conidial Germination and Morphological Changes in Germinated Conidia

Each natural compound was first liquefied in 10 µL of DMSO before being diluted with distilled water to a concentration of 0.1 µg/mL. As a positive control, a 0.1 µg/mL solution of Nativo^®^WG75 was prepared in distilled water. We used the methodology developed previously by us for MoT isolate conidial germination investigations [31,55,56]. Briefly, a 100 μL solution containing 1 × 10^5^ conidia/mL of BTJP 4 (5) was directly mixed with a 100 μL solution containing 0.1 μg/mL of product to obtain a 200 μL final solution in the well of a 96-multiwell plate containing test compounds comprising 0.5 μg/mL. Immediately after blending with a glass rod, the suspension was incubated for 6, 12, and 24 h at 25 °C in a Ziploc plastic bag with layers of moist paper towel. Sterile water that contained 1% DMSO was employed as a control. A total of 100 conidia from each of the five replications were examined with a Zeiss Primo Star microscope at a 100× magnification. The photographs were acquired with a Zeiss Axiocam ERc 5s, and the percentage of conidia germination, and the morphological alterations of spore germ tubes and appressoria, were determined. The experiment was repeated five times, with at least five replications for each treatment. The conidia germination percentage was calculated as: CG% = (C − T)/C × 100; where %CG = conidia germination, C = average conidia germination percentage in control, and T = average conidia germination percentage in treated samples. 

### 4.6. Wheat Blast Progression on Detached Wheat Leaves

Bonactin and feigrisolide C stock solutions were made using a small quantity of DMSO. The final DMSO content never exceeded 1% when the natural substances were dissolved in sterile distilled water to obtain concentrations of 1, 5, and 10 µg/mL. Nativo^®^WG75 was prepared in concentrations of 1, 5, and 10 µg/mL as well. As a negative control, sterilized water that contained 1% DMSO was utilized. This experiment was carried out according to the procedures outlined by Chakraborty et al. [31,55,56]. The first step was to separate wheat leaves from seedlings at the five-leaf stage and place them on plates covered with wet paper towels. Each leaf was treated with three 20 µL drops of the appropriately prepared test compound at the aforementioned concentrations, and the leaves were left to dry for 15 min. Following that, inoculation was conducted on each spot with 1 µL conidial solution containing 1 × 10^5^ BTJP 4 (5) conidia/mL, and the plates were incubated at 28 °C in the darkness for the first 30 h, then under constant lighting for the following two days. The experiment was repeated five times with five different samples each time. For each treatment and compound concentration, the diameter of blast lesions induced by MoT was measured on three leaves per experiment.

### 4.7. Determination of Wheat Blast Control Efficacy of Bonactin and Feigrisolide C under Field Conditions

#### 4.7.1. Soil Preparation and Seed Sowing

The experiment was carried out in the research field of the Bangabandhu Sheikh Mujibur Rahman Agricultural University (BSMRAU) in Gazipur, Bangladesh. The trial site was situated 8.4 m above sea level at a latitude of 24.09° north, and a longitude of 90.26° east. Weeds and stubbles were pulled out of the soil after it had been gently plowed. During soil preparation, adequate quantities of well-decomposed cow dung were applied. Gypsum, muriate of potash, triple super phosphate, and urea were applied as chemical fertilizers at a rate of 11-50-28-70 kg/ha. [63]. 3 to 4 days before seed sowing, the final soil preparation included the application of additional fertilizers along with two-thirds of the urea as a baseline dose. 20 days after the first irrigation, the final one-third of the urea was applied. In the first week of December, BARI Gom-26 wheat seeds were sown. Before sowing, the seeds were treated with Vitavex 200 (3 g/kg seed). There were three replications per treatment and the size of the experimental plot was 1 m^2^. All of the plots were properly labeled. The required irrigation work was performed, along with additional cross-cultural tasks. The experiment was conducted using a randomized complete block design (RCBD).

#### 4.7.2. Infection Assay in the Wheat Reproductive Phase 

The test compounds were applied at a concentration of 5 µg/mL in each plot and allowed to dry overnight, whereas sterile water containing 1% DMSO served as a negative control. BTJP 4 (5) spore suspension was sprayed to wheat fields immediately after flowering. The positive control was the fungicide Nativo^®^ 75WG, whereas the negative control was deionized distillate water. To establish a humid atmosphere suitable for spore germination, polyethylene sheets were placed over plots before inoculation.

#### 4.7.3. Data Collection and Analysis for Disease Severity

During the reproductive phase, data were recorded on the total number of tillers, productive tillers, infected tillers per hill, the full length of the spikes, diseased area of the spikes, seeds per spike, 1000-grain weight, and grain production per hill. During the vegetative phase, data were collected on the total number of seedlings, the number of infected seedlings per pot, the overall length of the leaves, and the infected area of the leaves. The disease intensity (DI) was calculated using the formula:
DI =Total infected plants× 100
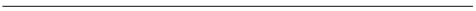
Total plants observed

A 5-point scale was used to assess the severity of blast disease, with % infection accounting for the length of the spike that was infected by blast. The scales were 0 for the absence of lesions, 1 for infection rates between 1% and 25%, 2 for infection rates between 26% and 50%, 3 for infection rates between 51% and 75%, and 4 for infection rates between 76% and 100% on the length of damaged leaves. Blast severity was measured by the following formula:
DS =*n* × v× 100%
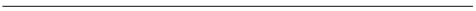
N × V

where DS = disease severity*n* = number of blast-infected leavesv = value score for blast severityN = number of observed leavesV = value of highest score.

### 4.8. Statistical Analysis, Experimental Design, and Replications

The efficacy of the pure compounds was examined in the laboratory and the field, respectively, using completely randomized design (CRD) and randomized complete block design (RCBD). All statistical analyses were performed using Microsoft Office Excel 2015 and IBM SPSS Statistics 25. Tukey’s HSD (honest significance difference) test was used to compare the treatment means. The tables and figures utilized the mean value ± standard error and there were five replications per treatment.

## 5. Conclusions

In this study, we demonstrated for the first time that marine natural products, bonactin and feigrisolide C, from *Streptomyces* species, suppressed the mycelial growth and asexual development of an isolate of MoT fungus and inhibited the progression of wheat blast disease caused by that isolate in vivo. Large-scale in vitro and field testing of these compounds with multiple isolates is necessary to determine whether they are potential candidates or lead compounds for developing an effective fungicide against wheat blast disease. More investigation is also needed to determine their level of toxicity towards humans and the environment, as well as their specific method of action and the structure–activity association between these bioactive natural compounds and the wheat blast fungus *M. oryzae Triticum*.

## Figures and Tables

**Figure 1 plants-11-02108-f001:**
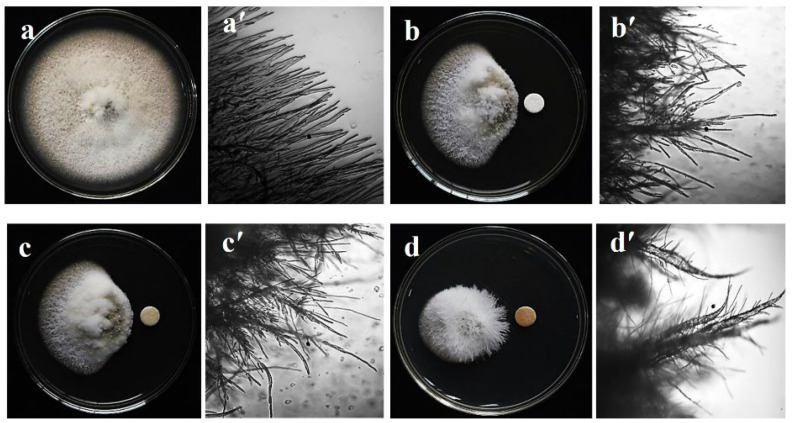
Mycelial growth suppression and morphological changes of hyphae of a wheat blast fungus, *Magnaporthe oryzae Triticum* (MoT) isolate BTJP 4 (5) approaching the paper disks containing two marine natural products, bonatin and feigrisolide C, and Nativo^®^ WG75 (20 µg/disk), a commercial fungicide known to growers as local standard in Bangladesh. Normal mycelial growth (**a**) of BTJP 4 (5) on PDA plate (10 days) and microscopic view of the growing typical tubular hyphal tips (**a′**) in the untreated control. Mycelial growth inhibition (**b**) and abnormal hyphal tips (**b′**) closer to the paper disk containing bonactin. Inhibited mycelia (**c**) and curly and irregular growth of hyphal tips (**c′**) by feigrisolide C. Mycelial growth inhibition (**d**) and severely damaged hyphal tips (**d′**) by the Nativo^®^ WG75. Bar = 50 μm. The micrographs shown in panels A and B were captured with a digital camera (CAMEDIA C-3040 zoom; Olympus Optical Co. Ltd., Tokyo, Japan), and those in panels C and D were taken from a light microscope (IX70-S1F2; Olympus) by using the same digital camera connected to it.

**Figure 2 plants-11-02108-f002:**
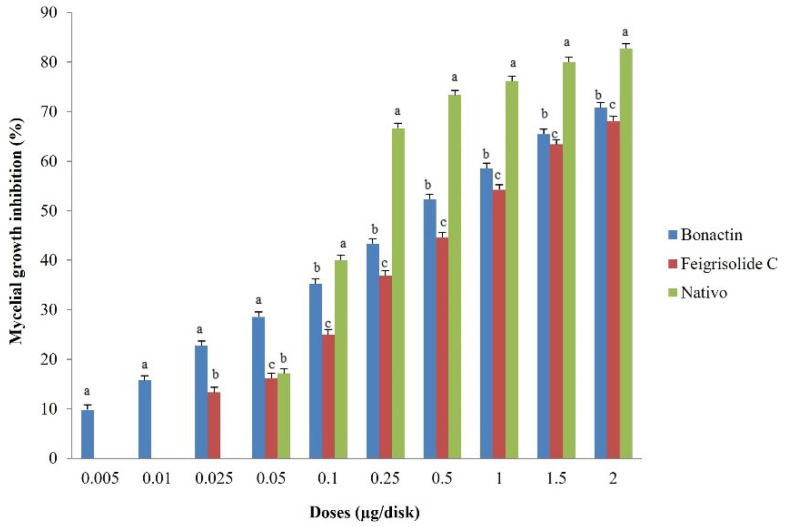
Suppression effects of bonactin, feigrisolide C, and Nativo^®^ WG75 on mycelial growth of *Magnaporthe oryzae Triticum* (MoT) isolate BTJP 4 (5) in PDA media. The data represents the mean ± standard errors of three replications for each rate of the test compound based on the Tukey HSD (honest significance difference) test at a 5% level.

**Figure 3 plants-11-02108-f003:**
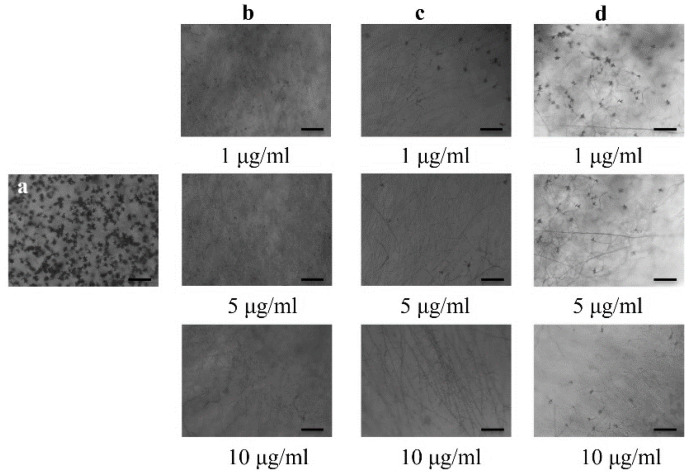
Effects of bonactin, feigrisolide C, and Nativo^®^ WG75 on suppression of conidiogenesis of *M. oryzae Triticum* isolate BTJP 4 (5) in the 96-multiwell plates at 1 μg/mL, 5 μg/mL, and 10 μg/mL. Image (**a**) control. Images in panels (**b**–**d**) are bonactin, feigrisolide C, and Nativo^®^ WG75, respectively. Bar = 50 μm.

**Figure 4 plants-11-02108-f004:**
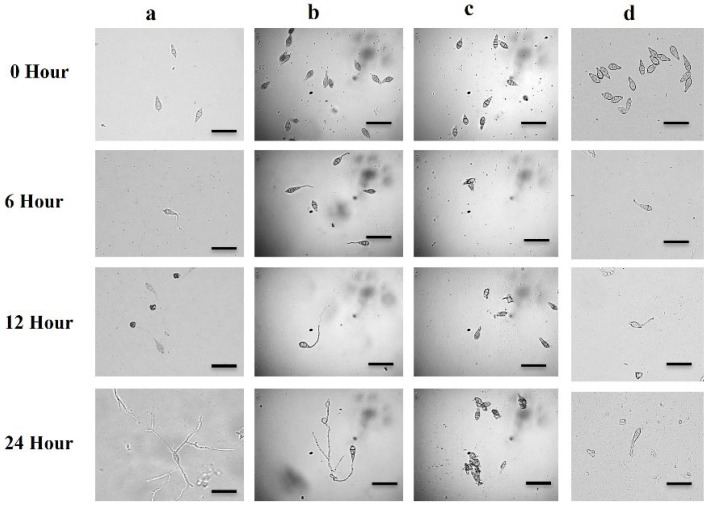
Micrographs showing the changes in germination and developmental transitions of MoT conidia with time-course in the untreated control (panel (**a**)) and the presence of bonactin (panel (**b**)), feigrisolide C (panel (**c**)), and a commercial fungicide Nativo^®^ WG75 (panel (**d**)) at 0.5 µg/mL. Bar = 10 μm.

**Figure 5 plants-11-02108-f005:**
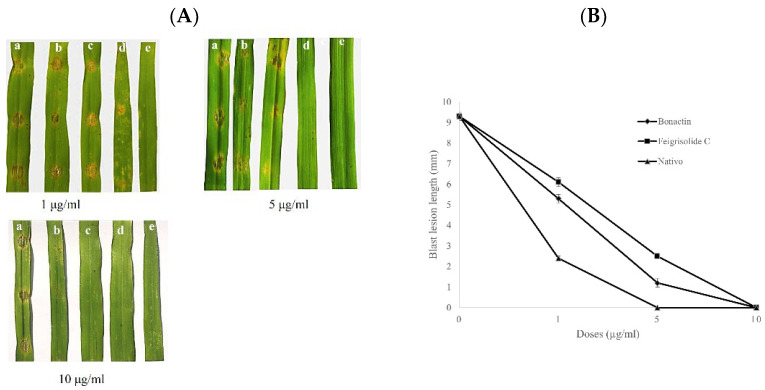
(**A**). Representative images showing wheat blast disease (symptoms) suppression by varying doses (1–10 μg/mL) of bonactin, feigrisolide C, and Nativo^®^ WG75. The compounds were liquefied in 1% DMSO and applied on the detached leaves of wheat (cv. BARI Gom 26) 24 h before artificial point inoculation with 20 µL/point of suspension of conidia containing 1 × 10^5^ conidia/mL. (a) Control, 1% DMSO, (b) bonatin, (c) feigrisolide C, (d) Nativo^®^ WG75, and (e) uninoculated and untreated leaf. (**B**) Average lengths of blast lesions on detached wheat leaves pretreated with bonatin, feigrisolide C, and Nativo^®^WG75 compared to 1% DMSO treatment control. The data are the means ± standard errors of at least five replications for each dosage of the compounds at *p* ≤ 0.05. Vertical bars represent ± standard error.

**Figure 6 plants-11-02108-f006:**
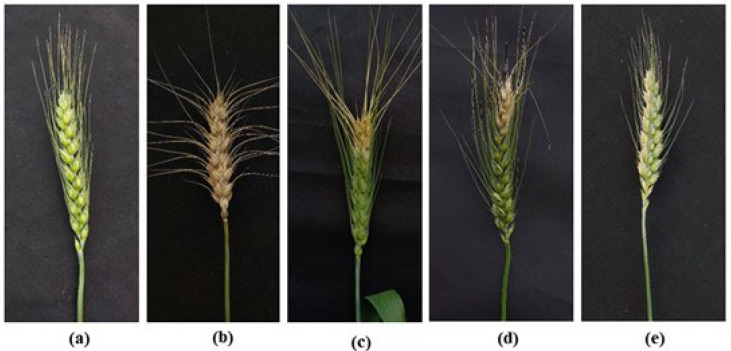
Inhibition of wheat blast disease with bonactin, feigrisolide C at 5 μg/mL, and Nativo^®^ 75WG at 50 μg/mL; (**a**) Uninoculated, untreated spike, (**b**) BTJP 4 (5) inoculation + water control + (**c**) bonactin + BTJP 4 (5) inoculation, (**d**) feigrisolide C + BTJP 4 (5) inoculation, (**e**) Nativo^®^75WG + BTJP 4 (5) inoculation.

**Figure 7 plants-11-02108-f007:**
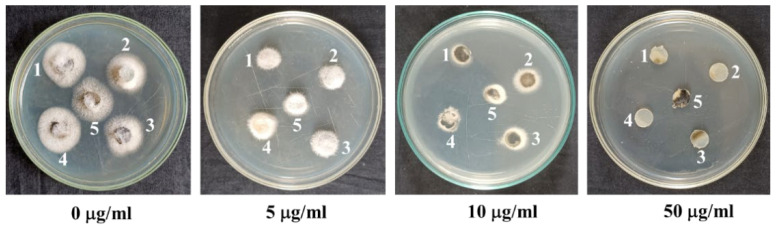
Sensitivity of different strains (**1**–**5**) of wheat blast fungus *Mahnaporthe oryzae Triticum* obtained from the field-infected spikes to various doses of a commercial fungicide Nativo. **1**, BTKP 22(3) collected in 2022; **2**, BTJP 194-2 collected in 2019; **3**, BTJP 1910-3 collected in 2019; **4**, BTJP 2 g collected in 2017; and **5**, BTJP 4 (5) collected in 2016. The PDA plates were cultured at 25 °C for 3 days after inoculation of the plates by various wheat blast strains.

**Figure 8 plants-11-02108-f008:**
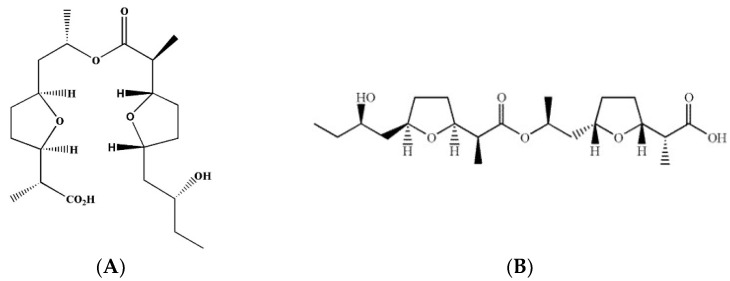
Structures of bonactin (**A**) and feigrisolide C (**B**).

**Table 1 plants-11-02108-t001:** In vitro effects of bonactin and feigrisolide C on conidia germination and the developmental transitions of *M. oryzae Triticum* (MoT) isolate BTJP 4 (5) at 0.5 µg/mL.

Compound	Time (h)	Effects of Secondary Metabolites on the Developmental Alterations of Conidia of a MoT Isolate	
		Germinated Conidia (% ± SE ^a^)	Major Morphological Changes Occurred in the Treated Conidia
Water	0	0 ± 0 ^e^	No germination
	6	100 ± 0 ^a^	Normal germ tube and development of normal appressoria
	12	100 ± 0 ^a^	Hyphal growth was observed
	24	100 ± 0 ^a^	Huge hyphal growth occurred
Bonactin	0	0 ± 0 ^e^	Zero germination
	6	79.1 ± 0.6 ^b^	Germinated conidia had short germ tube
	12	79.1 ± 0.6 ^b^	12.7 ± 0.4% Normal germ tube and 66.5 ± 0.5% of germ tube formed unusually elongated branches
	24	69.6 ± 0.5 ^b^	9.5 ± 0.2% Normal appressoria and 60.1 ± 0.3% abnormal appressoria (low melanization) but no hyphal growth
Feigrisolide C	0	0 ± 0 ^e^	No germination
	6	7.4 ± 0.5 ^d^	7.4 ± 0.5% conidia lysed; No germination took place
	12	0 ± 0 ^d^	No germination took place
	24	0 ± 0 ^c^	No germination took place
Nativo^®^ WG75	0	0 ± 0 ^e^	Zero germination
	6	49.7 ± 0.6 ^c^	Germinated, but germ tube was very short
	12	49.7 ± 0.6 ^c^	Normal germ tube formed
	24	0 ± 0 ^c^	Zero appressoria formed; zero hyphal growth

^a^ Data are mean value ± SE of three replications in each natural compound. Means within a column followed by a different letter(s) are significantly different according to Tukey’s HSD (honest significance difference) post-hoc (*p* ≤ 0.05).

**Table 2 plants-11-02108-t002:** Effect of bonactin and feigrisolide C on wheat (variety-BARI Gom-26) yield and yield components in field conditions following the artificial inoculation with BTJP 4 (5).

Treatment	Yield/1 m^2^ Plot (gm) *	1000-Grain Weight (gm) *	Blast Incidence (%) *	Blast Severity (%) *
Healthy control	133.07 ± 2.33a	46.63 ± 1.57a	0.00 ± 0.00e	0.00 ± 0.00d
Untreated control	64.60 ± 1.71c	31.77 ± 1.29c	87.33 ± 3.18a	82.67 ± 3.53a
Bonactin	112.97 ± 2.26b	40.09 ± 1.72ab	41.00 ± 1.15c	32.33 ± 2.40b
Feigrisolide C	106.40 ± 2.58b	38.78 ± 3.16b	51.33 ± 3.53b	38.67 ± 1.20b
Nativo^®^ 75WG	126.10 ± 2.70a	43.28 ± 2.52ab	24.00 ± 4.04d	14.33 ± 2.33c

* Yield data are the mean ± SE collected from five replications of each treatment of the test compounds. Data followed by the same letter in a column are not significantly different according to Tukey HSD (honest significance difference) post-hoc statistic at the 5% level.

## Data Availability

The manuscript includes all the data.

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
