# Peer review of "Bonactin and Feigrisolide C Inhibit Magnaporthe oryzae Triticum Fungus and Control Wheat Blast Disease"

_plants, 2022, doi:10.3390/plants11162108_

Round 1

Reviewer 1 Report

The current manuscript describes the inhibitory effect of two microbial secondary metabolites (described as bonactin and feigrisolide C) extracted/obtained from marine bacteria from the genus Streptomyces. The model system tested was the wheat fungal pathogen Magnaporthe (Pyricularia oryzae) Triticum lineage, which caused wheat blast. The final objective of this study was to check if these two compounds could control the wheat blast disease by reducing blast severity under controlled environment (incubator) and in field conditions.

The research would be relevant for improving the management of the wheat blast disease, specially for seeking an alternative sustainable solution for excessive fungicide sprays with limitted efficacy, in the absence of varietal resistance.

My first main concern about their study is the misleading assertion about the safity of the metabolites tested. It is clear that the authors associated distinct natural properties of the bioactive compounds by  comparing them with the conventional chemical fungicides. However, without proper toxicity dat and environmental impact information on these two actives, these clains are simply questionable once these microbial metabolites have antibiotic/antifungic properties. The authors, in fact, compare these two bioactives to agricultural antibiotics (such as kasugamycin, blasticidin S, validamycin, and polyoxin) used for managing Pyricularia oryzae on rice crops.  I would recommend the authors refrain from this line of thought, and consider the fact that these antibiotics are harmful and not totally safe for non-target organisms and to the environment due to their toxicity and their high risk for antimicrobial resistance selection associated with their specific (single target perhaps) mode of action.

It is simply misleading to make a clain that these bioactives might be safer than chemical fungicides just because they were naturally derived. Take, for example, the case of strobilurin fungicides, which were initially isolated from the wood-rotting mushroom Strobilurus tenacellus. It is now a chemically sinthesized fungicide with inherent toxicity and enviromental impact.  

The authors even stated: "When subjected to natural ecology,  often decay within a month or even a few days, leaving low residual rates that might be less harmful to natural ecosystems".  To contrapose this assertion, I will paraphrase EPA / US: "The terms “naturally” and “renewable” suggest that a particular product contains certain ingredients that are safer than other products that contain other ingredients. EPA does not approve claims that suggest a pesticide is safe, and does not approve claims that could be considered misleading comparative claims about the safety of a product versus other products that do not contain these same ingredients. LC08-0187; 7.8.08" .

At this point I would like to ask the authors why not testing the role of the  bonactin and feigrisolide C producing Streptomyces as biocontrol agents against wheat blast? Or even better, to approach the advantages / disvantagens /pitfalls of selecting microbial metabolites to develop an antibiotic for agricultural usage other than formulating the microrganism as biocontrol agent for agricultural deployment. 

The second main concern is that this study does not provide biological replicares of  the observations. Therefore the experimental design is only somewhat sound for testing the hypotheses proposed,  because all the conclusions were based on the reaction / screening of a single fungal isolate. The pathogen is well known for its high genetic and phenotypic diversity and a single monosporic isolate certainly is not representative for such important claims made in the manuscript.

If Plant does not require biological replicates (minimum of three fungal isolates) in their intented manuscripts, it is fine with me. But the authors should then soften their claims and stress that their experiments lack biological replication and therefore should be interpreted cautiosly. 

And if that is the case, the authors concluded that bonactin and feigrisolide C inhibited fungal mycelial development, conidia production, conidial germination, and morphological modifications in germinating conidia and suppressed wheat blast disease in vivo, and could be promising antibiotics / antifungic coumpounds that could be developed / tested for wheat blast management.

Author Response

Reviewer 1:

Open Review

English language and style

( ) Extensive editing of English language and style required
(x) Moderate English changes required
( ) English language and style are fine/minor spell check required
( ) I don't feel qualified to judge about the English language and style

Our responses:

Thank you for your suggestion. We have revised the manuscript.

Yes

Can be improved

Must be improved

Not applicable

Does the introduction provide sufficient background and include all relevant references?

(x)

( )

( )

( )

Are all the cited references relevant to the research?

(x)

( )

( )

( )

Is the research design appropriate?

( )

( )

(x)

( )

Are the methods adequately described?

( )

(x)

( )

( )

Are the results clearly presented?

(x)

( )

( )

( )

Are the conclusions supported by the results?

( )

(x)

( )

( )

Comments and Suggestions for Authors

The current manuscript describes the inhibitory effect of two microbial secondary metabolites (described as bonactin and feigrisolide C) extracted/obtained from marine bacteria from the genus Streptomyces. The model system tested was the wheat fungal pathogen Magnaporthe (Pyricularia oryzae) Triticum lineage, which caused wheat blast. The final objective of this study was to check if these two compounds could control the wheat blast disease by reducing blast severity under controlled environment (incubator) and in field conditions. The research would be relevant for improving the management of the wheat blast disease, specially for seeking an alternative sustainable solution for excessive fungicide sprays with limited efficacy, in the absence of varietal resistance.

Our response:

Many thanks for your kind comment and summarizing our work so precisely.

My first main concern about their study is the misleading assertion about the safety of the metabolites tested. It is clear that the authors associated distinct natural properties of the bioactive compounds by comparing them with the conventional chemical fungicides. However, without proper toxicity dat and environmental impact information on these two actives, these claims are simply questionable once these microbial metabolites have antibiotic/antifungic properties. The authors, in fact, compare these two bioactives to agricultural antibiotics (such as kasugamycin, blasticidin S, validamycin, and polyoxin) used for managing Pyricularia oryzae on rice crops.  I would recommend the authors refrain from this line of thought and consider the fact that these antibiotics are harmful and not totally safe for non-target organisms and to the environment due to their toxicity and their high risk for antimicrobial resistance selection associated with their specific (single target perhaps) mode of action.

It is simply misleading to make a claim that these bioactives might be safer than chemical fungicides just because they were naturally derived. Take, for example, the case of strobilurin fungicides, which were initially isolated from the wood-rotting mushroom Strobilurus tenacellus. It is now a chemically synthesized fungicide with inherent toxicity and environmental impact.  

Our responses:

Thank you for the comment. We understand your valid concern, but very few studies have been conducted regarding the toxicity of these compounds. Tang et al. (2000) and Sobolevskaya et al. (2004) have found that Feigrisolide C had least toxicity against tumor cells, L-929 (mouse flbroblasts), K562 (human leukemia), and HeLa (human cervix carcinoma). However, Feigrisolide C does not involve in destruction of the biological membranes and do not possess membranotropic activity and induces apoptotic process (Sobolevskaya et al. 2004; Prokofeva et al. 1996). Bonactin is not carcinogenic and is not toxic to aquatic model organisms. It has already been reported as a suitable natural compound against the Schizophrenia disorder (Thiyagarajamoorthy et al. 2018). We didn’t find any report regarding their toxicity towards humans and environment. As they are naturally derived product, we have suggested them as safer than synthetic fungicides. However, we can’t make any prediction at this time pertaining to the toxicity if these compounds are chemically synthesized to use in formulated fungicide products. We have revised the sentence for avoiding confusion.  

References:

Tang, Y.-Q.; Sattler, I.; Thiericke, R.; Grabley, S.; Feng, X.-Z. Feigrisolides A, B, C and D, new lactones with antibacterial activities from Streptomyces griseus. J. Antibiot. 2000, 53, 934–943.

Sobolevskaya, M.P.; Fotso, S.; Havash, U.; Denisenko, V.A.; Helmke, E.; Prokofeva, N.G.; Kuznetsova, T.A.; Laatsch, H.; Elyakov, G.B. Metabolites of the sea isolate of bacteria Streptomyces sp. 6167. Chem. Nat. Comp. 2004, 40, 282–285.

Thiyagarajamoorthy, D.K., Arulanandam, C.D., Dahms, HU. et al. Marine Bacterial Compounds Evaluated by In Silico Studies as Antipsychotic Drugs Against Schizophrenia. Mar Biotechnol 20, 639–653 (2018). https://doi.org/10.1007/s10126-018-9835-3

Prokofeva, N.G., Kalinovskaya, N.I., Lukyanov, P.A., Kuznetsova, T.А. Membranotropic effects of cyclic lipopeptides produced by a marine isolate of the bacteria Bacillus pumilus, Biologiya morya, 22 (3), 167-170 (1996).

The authors even stated: "When subjected to natural ecology, often decay within a month or even a few days, leaving low residual rates that might be less harmful to natural ecosystems".  To contrapose this assertion, I will paraphrase EPA / US: "The terms “naturally” and “renewable” suggest that a particular product contains certain ingredients that are safer than other products that contain other ingredients. EPA does not approve claims that suggest a pesticide is safe and does not approve claims that could be considered misleading comparative claims about the safety of a product versus other products that do not contain these same ingredients. LC08-0187; 7.8.08”.

Our responses:

Thank you for the comment. We made this statement regarding the importance of using natural secondary metabolites compared to that of synthetic fungicides. It has been reported that microorganisms derived metabolites are easily degraded in the environment (Tanaka and Omura, 1993). As these compounds are the precursors of nonactic acid esters, and nonactic acid esters are reported as environmentally benign since soil microbes convert them to H2O and CO2 (Zizka 1998), we have suggested them as  possible lead components for the synthesis of agricultural fungicides toward MoT. Once again, we do not know if pesticide companies will consider these compounds as new active ingredients and if that happen what will be the toxicity level of those products. Our statemnents are analogous to some of the findings from a very limited number of studies.

Reference:

Tanaka, Y.T.; Omura, S. Agroactive compounds of microbial origin. Annu. Rev. Microbiol. 1993, 47, 57–87.

Zizka, Z. Biological effects of macrotetrolide antibiotics and nonactic acids. Folia Microbiol. 1998, 43, 7–14.

At this point I would like to ask the authors why not testing the role of the bonactin and feigrisolide C producing Streptomyces as biocontrol agents against wheat blast? Or even better, to approach the advantages / disadvantages /pitfalls of selecting microbial metabolites to develop an antibiotic for agricultural usage other than formulating the microorganism as biocontrol agent for agricultural deployment. 

Our responses:

Thank you for the comment. We did not use Streptomyces strain as biocontrol agents against wheat blast in this study. However, we have good experiences on this issue. Our main goal of this study is to evaluate the antifungal performance of bonactin and feigrisolide C  under in vitro and in vivo condition that could be used as  lead compounds to formulate a new products/fungicides. We have a plan to conduct experiment with Streptomyces strain in our next project.

The second main concern is that this study does not provide biological replicates of the observations. Therefore, the experimental design is only somewhat sound for testing the hypotheses proposed, because all the conclusions were based on the reaction / screening of a single fungal isolate. The pathogen is well known for its high genetic and phenotypic diversity and a single monosporic isolate certainly is not representative for such important claims made in the manuscript.

If Plant does not require biological replicates (minimum of three fungal isolates) in their intended manuscripts, it is fine with me. But the authors should then soften their claims and stress that their experiments lack biological replication and therefore should be interpreted cautiously. 

And if that is the case, the authors concluded that bonactin and feigrisolide C inhibited fungal mycelial development, conidia production, conidial germination, and morphological modifications in germinating conidia and suppressed wheat blast disease in vivo and could be promising antibiotics / antifungic compounds that could be developed / tested for wheat blast management.

Our responses:

Thank you for the comment. As we mentioned in our manuscript, we have isolated MoT strain BTJP 4 (5) in 2016 from the blast-infected wheat field from Jhenaidah which is a hot spot for wheat blast in Bangladesh where highest yield losses were recorded (51%) (Islam et al. 2016). Among other isolates of our collection from other wheat blast affected regions, we have observed that this isolate is the most aggressive, vigorously growing and damaging one. We have tested these compounds against all other isolates and got same inhibitory activities as the introduced MoT population in Bangladesh was a clonal. Please find new data on equivalent sensitivities of the MoT isolates collected in different years after the first epidemic of wheat blast in Bangladesh in 2016 toward Nativo fungicide. Considering the aggressiveness of this strain over other isolates, we have decided to report the suppressive effect of  these compounds against it. Regarding your concern, we have revised our statement as you suggested. We have already reported several compounds and nanoparticles against this strain, and they have been published in different reputed journals, which are listed below:

References:

Chakraborty, M.; Mahmud, N.U.; Gupta, D.R.; Tareq, F.S.; Shin, H.J.; Islam, T. Inhibitory effects of linear lipopeptides from a marine Bacillus subtilis on the wheat blast fungus Magnaporthe oryzae Triticum. Front. Microbiol. 2020, 11, 665. https://doi.org/10.3389/fmicb.2020.00665

Chakraborty, M.; Mahmud, N.; Muzahid, A.N.M.; Rabby, S.M.F.; Islam, T. Oligomycins inhibit Magnaporthe oryzae Triticum and suppress wheat blast disease. PLoS ONE 2020, 15, e0233665. https://doi.org/10.1371/journal.pone.0233665

Paul, S.K.; Chakraborty, M.; Rahman, M.; Gupta, D. R.; Mahmud, N.U.; Rahat, A.A.M; Sarker, A.; Hannan, M.A.; Rahman, M.M.; Akanda, A.M.; Ahmed, J.U.; Islam, T. Marine Natural Product Antimycin A Suppresses Wheat Blast Disease Caused by Magnaporthe oryzae Triticum . J. Fungi (Acceptance date: 06/06/2022).

Mahmud, N.U., Gupta, D.R., Paul, S.K., Chakraborty, M., Mehebub, M.S., Surovy, M.Z., Rabby, S.F., Rahat, A.A.M., Roy, P.C., Sohrawardy, H. and Amin, M.A., 2022. Daylight-Driven Rechargeable TiO2 Nanocatalysts Suppress Wheat Blast Caused by Magnaporthe oryzae Triticum. Bulletin of the Chemical Society of Japan. https://doi.org/10.1246/bcsj.20220010

Islam, M.T.; Croll, D.; Gladieux, P.; Soanes, D.M.; Persoons, A.; Bhattacharjee, P.; Hossain, M.; Gupta, D.R.; Rahman, M.; Mahboob, M.G.; Cook, N. Emergence of wheat blast in Bangladesh was caused by a South American lineage of Magnaporthe oryzae. BMC biol. 2016, 14, 1-1.  

Reviewer 2 Report

Wheat blast caused by Magnaporthe oryzae Triticum (MoT) pathotype causing significant yield losses. Limited numbers of fungicides are effective against the pathogen and further occurrence of fungicide-resistant in pathogens made situation more complex. Use of synthetic fungicides also causes adverse environmental affects and hazards to human health. Therefore, the bioactive compound from naturally occurring organism may be useful in managing this problem. The manuscript is well written but need clarification on following points

In the methodology it is not clear that how the stock solution has been prepared ‘Stock solutions of every natural compound were made by liquefying 0.1 μg compound in 10 μl DMSO the slock solutions” How some small quantities of 0.1 μg are weighed, need more clarity to follow the methods to other laboratories.  

In lab experiment compounds including Nativo found very effective at very low conc. 1, 5 and 10 μg/ml, even in conidia germination inhibition by feigrisolide C cause 100% inhibition at 0.05 μg/ml conc.

In the field experiment the field or plot size, numbers of replications etc are not mentioned whereas yield is given per plot basis.

Under the result section “a field experiment was conducted with a commercial fungicide, Nativo®75WG at 5 g/ml” why so higher doses of the chemical has been taken? The recommendation is quite lower of Nativo or in laboratory the compounds are effective at ultra low conc. in comparison of the conc. taken in field.

Author Response

Reviewer 2:

Open Review

English language and style

( ) Extensive editing of English language and style required
( ) Moderate English changes required
(x) English language and style are fine/minor spell check required
( ) I don't feel qualified to judge about the English language and style

Our responses:

Thank you for the comment. We have revised the manuscript.

Yes

Can be improved

Must be improved

Not applicable

Does the introduction provide sufficient background and include all relevant references?

(x)

( )

( )

( )

Are all the cited references relevant to the research?

(x)

( )

( )

( )

Is the research design appropriate?

(x)

( )

( )

( )

Are the methods adequately described?

( )

(x)

( )

( )

Are the results clearly presented?

(x)

( )

( )

( )

Are the conclusions supported by the results?

(x)

( )

( )

( )

Comments and Suggestions for Authors

Wheat blast caused by Magnaporthe oryzae Triticum (MoT) pathotype causing significant yield losses. Limited numbers of fungicides are effective against the pathogen and further occurrence of fungicide-resistant in pathogens made situation more complex. Use of synthetic fungicides also causes adverse environmental affects and hazards to human health. Therefore, the bioactive compound from naturally occurring organism may be useful in managing this problem.

Our responses:

Thank you for precisely summarizing our work.

The manuscript is well written but need clarification on following points.

Our responses:

Many thanks for the encouraging comments and suggestions. We have revised the manuscript accordingly.

In the methodology it is not clear that how the stock solution has been prepared ‘Stock solutions of every natural compound were made by liquefying 0.1 μg compound in 10 μl DMSO the slock solutions” How some small quantities of 0.1 μg are weighed, need more clarity to follow the methods to other laboratories. 

Our responses:

Thank you for the comment. We actually prepared the stock solutions of each compound by liquefying in 10 μl of DMSO and then diluted with distilled water to make the concentration down to 0.1 µg/ml. We have revised the sentence for better understanding.

In the field experiment the field or plot size, numbers of replications etc are not mentioned whereas yield is given per plot basis.

Our responses:

Thank you for the comment. In the field experiment, the plot size was 1m2 and each treatment was replicated three times for the field study. We have added this information in the manuscript.

In lab experiment compounds including Nativo found very effective at very low conc. 1, 5 and 10 μg/ml, even in conidia germination inhibition by feigrisolide C cause 100% inhibition at 0.05 μg/ml conc. Under the result section “a field experiment was conducted with a commercial fungicide, Nativo®75WG at 5 g/ml” why so higher doses of the chemical have been taken? The recommendation is quite lower of Nativo or in laboratory the compounds are effective at ultra-low conc. in comparison of the conc. taken in field.

Our responses:

Many thanks for the comment and pointing out a major editing mistake. We actually used 5 µg/mL of all the compounds including the 50 µg/mL Nativo®75WG (this concentration completely inhibit the growth of MoT in vitro, please see new data added) for field experiment which is a lower dose than the recommended. The recommended commercial dose of Nativo®75WG is 0.6 gm/mL. We have revised it in the manuscript.

Round 2

Reviewer 1 Report

The comments made in the first round of reviews HAVE NOT BEEN ADDRESSED properly IN THE MANUSCRIPT itself, as expected, so my recommendation is to implement a major revision as suggested.

Those are the points that the authors should have approached IN the manuscript, but have not. ( Please do not worry about answering this reviewer. Rather, fix the manuscript according to the suggestions):

1. The authors compared the two bioactives (described as bonactin and feigrisolide C) to agricultural antibiotics (such as kasugamycin, blasticidin S, validamycin, and polyoxin) used for managing Pyricularia oryzae on rice crops. Refrain from this line of thought and consider the fact that these antibiotics are harmful and not totally safe for non-target organisms and to the environment due to their toxicity and their high risk for antimicrobial resistance selection associated with their specific (single target perhaps) mode of action.

2) On the response letter the authors pointed out: "We didn’t find any report regarding their toxicity towards humans and environment. As they are naturally derived product, we have suggested them as safer than synthetic fungicides." It was simply misleading to make a claim that these bioactives might be safer than chemical fungicides just because they were naturally derived. As mentioned earlier, take, for example, the case of strobilurin fungicides, which were initially isolated from the wood-rotting mushroom Strobilurus tenacellus. It is now a chemically synthesized fungicide with inherent toxicity and environmental impact.  Approach this IN the manuscript, as no changes have been made whatsoever. Use (incorporate) the statements from the response letter into your manuscript, if you wish. You were not asked to make predications, but you were asked to soften your statements about the safety these compounds without providing toxicity data towards humans and environment, and without any scientific support.

3) The authors kept exactly the same statement:  "When subjected to natural ecology, often decay within a month or even a few days, leaving low residual rates that might be less harmful to natural ecosystems". Paraphrasing EPA / US: "The terms “naturally” and “renewable” suggest that a particular product contains certain ingredients that are safer than other products that contain other ingredients. EPA does not approve claims that suggest a pesticide is safe and does not approve claims that could be considered misleading comparative claims about the safety of a product versus other products that do not contain these same ingredients. LC08-0187; 7.8.08”. A scientific journal should not approave such claims without scientific support, either. Please fix your statement accordingly, as suggested since analogous and not accurate statements continue to be not accurate. In view of scientific support, do not save words to support such claims, though. You provided several citations in the response letter. Incorporate them in the manuscript to support your claims. Abuse the "might be". Save the "should be".

4) The authors did not approach, in the manuscript, the advantages / disadvantages /pitfalls of selecting microbial metabolites to develop an antibiotic for agricultural usage other than formulating the microorganism as biocontrol agent for agricultural deployment, such as the role of the bonactin and feigrisolide C producing Streptomyces as biocontrol agents against wheat blast.

5) The other major concern was that this study did not provide biological replicates of the observations. Therefore, the experimental design was only somewhat sound for testing the hypotheses proposed, because all the conclusions were based on the reaction / screening of a single fungal isolate. The pathogen is well known for its high genetic and phenotypic diversity and a single monosporic isolate certainly is not representative for such important claims made in the manuscript. The fact that the authors claimed that the population of this organism (is) clonal (remain clonal after six years since its introduction in Bangladesh) does not justify the absence of biological replicates. /s mentioned. The authors also  stated in their response letter: "We have already reported several compounds and nanoparticles against this strain, and they have been published in different reputed journals, which are listed below".  Pardon to disagree with this statement, but claims such as these derived from a single strain is limited because it did not provide support from biological replicates, neither in Agronomic Sciences nor in the Medical Sciences. Even if Plant does not require biological replicates (minimum of three fungal isolates) IN their manuscripts, for the sake of science, the authors should then SOFTEN their claims and STRESS that their experiments lack biological replication and therefore should be interpreted CAUSTIOUSLY. This is simply assuming the limitations of the study.

6. I missed from indicating this problem in the first round of review, but for the most variables presented, there is not a single statistical test that could indicate significance differences among treatments, except for the data presented in Table 2. Even so, the author did not present results from the F tests and the sort of means comparisons that where implemented (Tukey, Duncan, Scott-Knot). Please provide this information.

7. The manuscript also requires extensive English style editing before acceptance, to either British and American English style. MDPI can sure provide such service.

These suggestions should be implemented in the manuscript, before recommending acceptance for publication.

Author Response

Our Responses to the Comments of the Reviewer 1:

Open Review

English language and style

( ) Extensive editing of English language and style required
(x) Moderate English changes required
( ) English language and style are fine/minor spell check required
( ) I don't feel qualified to judge about the English language and style

Our responses:

Thank you for your comments. We have revised the manuscript according to your suggestion.

Yes

Can be improved

Must be improved

Not applicable

Does the introduction provide sufficient background and include all relevant references?

(x)

( )

( )

( )

Are all the cited references relevant to the research?

(x)

( )

( )

( )

Is the research design appropriate?

( )

( )

(x)

( )

Are the methods adequately described?

( )

(x)

( )

( )

Are the results clearly presented?

(x)

( )

( )

( )

Are the conclusions supported by the results?

( )

(x)

( )

( )

Comments and Suggestions for Authors

The comments made in the first round of reviews HAVE NOT BEEN ADDRESSED properly IN THE MANUSCRIPT itself, as expected, so my recommendation is to implement a major revision as suggested.

Those are the points that the authors should have approached IN the manuscript but have not. (Please do not worry about answering this reviewer. Rather, fix the manuscript according to the suggestions):

  1. The authors compared the two bioactives (described as bonactin and feigrisolide C) to agricultural antibiotics (such as kasugamycin, blasticidin S, validamycin, and polyoxin) used for managing Pyricularia oryzae on rice crops. Refrain from this line of thought and consider the fact that these antibiotics are harmful and not totally safe for non-target organisms and to the environment due to their toxicity and their high risk for antimicrobial resistance selection associated with their specific (single target perhaps) mode of action.

Our responses:

Thank you for your concern. We have removed the misleading information from the manuscript.

2) In the response letter the authors pointed out: "We didn’t find any report regarding their toxicity towards humans and environment. As they are naturally derived products, we have suggested them as safer than synthetic fungicides." It was simply misleading to make a claim that these bioactives might be safer than chemical fungicides just because they were naturally derived. As mentioned earlier, take, for example, the case of strobilurin fungicides, which were initially isolated from the wood-rotting mushroom Strobilurus tenacellus. It is now a chemically synthesized fungicide with inherent toxicity and environmental impact.  Approach this IN the manuscript, as no changes have been made whatsoever. Use (incorporate) the statements from the response letter into your manuscript, if you wish. You were not asked to make predictions, but you were asked to soften your statements about the safety of these compounds without providing toxicity data towards humans and the environment, and without any scientific support.

Our responses:

Thank you for your kind suggestion. We are extremely sorry for the misconception. We have softened our statement and suggested further research before using these compounds for controlling wheat blast.

3) The authors kept exactly the same statement:  "When subjected to natural ecology, often decay within a month or even a few days, leaving low residual rates that might be less harmful to natural ecosystems". Paraphrasing EPA / US: "The terms “naturally” and “renewable” suggest that a particular product contains certain ingredients that are safer than other products that contain other ingredients. EPA does not approve claims that suggest a pesticide is safe and does not approve claims that could be considered misleading comparative claims about the safety of a product versus other products that do not contain these same ingredients. LC08-0187; 7.8.08”. A scientific journal should not approve such claims without scientific support, either. Please fix your statement accordingly, as suggested since analogous and not accurate statements continue to be not accurate. In view of scientific support, do not save words to support such claims, though. You provided several citations in the response letter. Incorporate them in the manuscript to support your claims. Abuse the "might be". Save the "should be".

Our responses:

Thank you for your suggestion. We have incorporated the citations in the manuscript and revised the statement for removing misleading information.

4) The authors did not approach, in the manuscript, the advantages / disadvantages /pitfalls of selecting microbial metabolites to develop an antibiotic for agricultural usage other than formulating the microorganism as biocontrol agent for agricultural deployment, such as the role of the bonactin and feigrisolide C producing Streptomyces as biocontrol agents against wheat blast.

Our responses:

Thank you for your suggestion. We have added the advantages of using microbial metabolites to develop an antibiotic for agricultural usage other than formulating the microorganism as a biocontrol agent for agricultural deployment in the manuscript.

5) The other major concern was that this study did not provide biological replicates of the observations. Therefore, the experimental design was only somewhat sound for testing the hypotheses proposed, because all the conclusions were based on the reaction / screening of a single fungal isolate. The pathogen is well known for its high genetic and phenotypic diversity and a single monosporic isolate certainly is not representative for such important claims made in the manuscript. The fact that the authors claimed that the population of this organism (is) clonal (remain clonal after six years since its introduction in Bangladesh) does not justify the absence of biological replicates. /s mentioned. The authors also stated in their response letter: "We have already reported several compounds and nanoparticles against this strain, and they have been published in different reputed journals, which are listed below".  Pardon to disagree with this statement but claims such as these derived from a single strain is limited because it did not provide support from biological replicates, neither in Agronomic Sciences nor in the Medical Sciences. Even if Plant does not require biological replicates (minimum of three fungal isolates) IN their manuscripts, for the sake of science, the authors should then SOFTEN their claims and STRESS that their experiments lack biological replication and therefore should be interpreted CAUSTIOUSLY. This is simply assuming the limitations of the study.

Our responses:

Thank you for your suggestion. We have revised our claims and suggested checking the efficacy of these compounds towards other MoT strains before using them as agrochemicals.

  1. I missed from indicating this problem in the first round of review, but for the most variables presented, there is not a single statistical test that could indicate significance differences among treatments, except for the data presented in Table 2. Even so, the author did not present results from the F tests and the sort of means comparisons that where implemented (Tukey, Duncan, Scott-Knot). Please provide this information.

Our responses:

Thank you for pointing out the editing errors. We have added the information and revised all the data accordingly.

  1. The manuscript also requires extensive English style editing before acceptance, to either British or American English style. MDPI can sure provide such service.

These suggestions should be implemented in the manuscript, before recommending acceptance for publication.

Our responses:

Thank you for your suggestion. We have revised our manuscript accordingly. We sincerely hope that this updated version o

Round 3

Reviewer 1 Report

I am impressed with the speed and the thoroughness of the authors in fixing the manuscript according to the reviewers´ many suggestions. This revealed high scientific profissionalism. The English editing was also well done. Therefore, my final recommendation is for its acceptance in this current form.